# Beneficial Effects of White Grape Pomace in Experimental Dexamethasone-Induced Hypertension

**DOI:** 10.3390/diseases13050132

**Published:** 2025-04-24

**Authors:** Raluca Maria Pop, Paul-Mihai Boarescu, Corina Ioana Bocsan, Mădălina Luciana Gherman, Veronica Sanda Chedea, Elena-Mihaela Jianu, Ștefan Horia Roșian, Ioana Boarescu, Floricuța Ranga, Maria Doinița Muntean, Maria Comșa, Sebastian Armean, Ana Uifălean, Alina Elena Pârvu, Anca Dana Buzoianu

**Affiliations:** 1Pharmacology, Toxicology and Clinical Pharmacology, Department of Morphofunctional Sciences, “Iuliu Haţieganu” University of Medicine and Pharmacy, Victor Babeș, No. 8, 400012 Cluj-Napoca, Romania; raluca.pop@umfcluj.ro (R.M.P.); abuzoianu@umfcluj.ro (A.D.B.); 2Department of Biomedical Sciences, Faculty of Medicine and Biological Sciences, “Stefan cel Mare” University of Suceava, 720229 Suceava, Romania; 3Clinical Emergency County Hospital Saint John the New, 720229 Suceava, Romania; 4Experimental Centre of “Iuliu Haţieganu” University of Medicine and Pharmacy, Louis Pasteur, No. 6, 400349 Cluj-Napoca, Romania; 5Research Station for Viticulture and Enology Blaj (SCDVV Blaj), 515400 Blaj, Romania; chedeaveronica@yahoo.com (V.S.C.);; 6Histology, Department of Morphofunctional Sciences, “Iuliu Haţieganu” University of Medicine and Pharmacy, Victor Babeș, No. 8, 400012 Cluj-Napoca, Romania; 7Niculae Stăncioiu Heart Institute Cluj-Napoca, 19–21 Calea Moților Street, 400001 Cluj-Napoca, Romania; dr.rosianu@gmail.com; 8Department of Cardiology—Heart Institute, “Iuliu Haţieganu” University of Medicine and Pharmacy, Calea Moților Street No. 19–21, 400001 Cluj-Napoca, Romania; 9Food Science and Technology, Department of Food Science, University of Agricultural Science and Veterinary Medicine Cluj-Napoca, Calea Mănăștur, No. 3–5, 400372 Cluj-Napoca, Romania; 10Pathophysiology, Department of Morphofunctional Sciences, Faculty of Medicine, “Iuliu Haţieganu” University of Medicine and Pharmacy, 400012 Cluj-Napoca, Romania; uifaleanana@gmail.com (A.U.); parvualinaelena@umfcluj.ro (A.E.P.)

**Keywords:** white grape pomace, antioxidant, antioxidant, dexamethasone-induced hypertension

## Abstract

Background: Grape pomace (GP), a by-product of winemaking, is a rich source of bioactive polyphenols known for their antioxidant and anti-inflammatory properties. While the cardiovascular benefits of red grape pomace have received significant scientific attention, the therapeutic potential of white grape pomace remains largely unexplored, particularly in glucocorticoid-induced hypertension. Given the rising prevalence of hypertension and the oxidative-inflammatory mechanisms underlying its progression, this study investigates the effects of white GP on blood pressure regulation, oxidative stress, and pro-inflammatory cytokine expression in an experimental model of dexamethasone (DEXA)-induced hypertension (HTN). By focusing on white GP, this research addresses a significant gap in current knowledge and proposes a novel, sustainable approach to managing hypertension through valorising winemaking by-products. Methods: The first concentration used, GP1, was 795 mg polyphenols/kg bw, while the second concentration, GP2, was 397.5 mg polyphenols/kg bw. Results: White GP polyphenols extract in the DEXA_GP1 group had reduced systolic and diastolic blood pressure. The extract with a higher content of polyphenols (GP1) prevented the elevation of serum levels of total oxidative stress (TOS), malondialdehyde (MDA), and oxidative stress index (OSI), while the extract with a lower content of polyphenols (GP2) slightly reduced serum levels of MDA. Both concentrations of GP increased serum levels of NO and Total Thiols, significantly higher (*p* < 0.05) than in the group treated with lisinopril. The serum levels of tumour necrosis factor-alpha (TNF-α) increased in all groups where HTN was induced. Both doses of GP extract prevented the elevation of TNF-α. Heart tissue levels of the studied cytokines (TNF-α, interleukin (IL)-1β, and IL-6 were not influenced (*p* > 0.05) by either the HTN induction or the treatment administered. Conclusions: These findings suggest that grape pomace may serve as a promising nutraceutical intervention for hypertension management, particularly in conditions associated with oxidative stress.

## 1. Introduction

Hypertension represents a key risk factor for ischemic heart disease, stroke, dementia, and chronic kidney disease. Globally, high blood pressure remains one of the most preventable causes of cardiovascular mortality and disease burden [1]. It is frequently linked to elevated oxidative stress and persistent inflammation, which contribute to vascular dysfunction and damage to target organs [2].

Dexamethasone (DEX), a synthetic glucocorticoid, is widely utilised in clinical practice to manage various medical conditions. It serves as a glucocorticoid hormone replacement in adrenal insufficiency, is combined with other chemotherapeutic agents for multiple myeloma treatment, reduces cerebral oedema caused by intracranial tumours, and acts as an anti-emetic for cancer patients [3,4,5,6]. However, chronic administration of DEX at high doses, which are not physiological, can lead to the development of hypertension [7].

Experimental models, such as dexamethasone-induced hypertension, mimic this pathophysiological interplay, providing insights into the mechanisms of hypertensive disease and potential therapeutic interventions [7].

The beneficial effects of natural dietary compounds have gained growing interest [8,9,10]. Many nutritional antioxidants are abundantly found in vegetables, seeds, and fruits [11]. Furthermore, utilising these antioxidants as a therapeutic approach is a well-established practice in both traditional and alternative medicine [12].

White grape pomace (GP), a by-product of winemaking, which is the solid remains of grapes after pressing for juice or wine, is rich in bioactive compounds, such as phenolic acids, flavonoids, and flavonols, which are known for their potent antioxidant and anti-inflammatory properties [13,14]. These compounds neutralise reactive oxygen species (ROS), enhance endogenous antioxidant systems, and modulate inflammatory pathways [15,16,17]. In the context of hypertension, GP has shown promise in mitigating oxidative stress and inflammation, thereby preserving vascular integrity and normalising blood pressure [18,19,20,21].

Most of the studies focus on evaluating the beneficial effects of grape seed extract [22]. Grape seed extract was proven to significantly lower systolic blood pressure and improve endothelial function in hypertensive rats by enhancing nitric oxide (NO) bioavailability [23]. Moreover, grape pomace from wine industrial waste was observed to produce relaxation of aortic rings, indicating that it might have vasodilatory effects mediated through the activation of endothelial nitric oxide synthase (eNOS) [24].

White grape pomace extract was shown to mitigate oxidative damage, suggesting its utility in reducing myocardial injury associated with ischemic events [11]. The antioxidant effect was attributed to the abundance of flavonoids, resveratrol, and tannins in the grape pomace, which neutralise free radicals, making it a valuable food supplement for supporting cardiovascular health [14].

Polyphenols from red grape products have been shown to modulate inflammatory pathways by downregulating pro-inflammatory cytokines, such as tumour necrosis factor (TNF)-α, interleukin (IL)-1β, IL-6 and IL-8, as a result of the inactivation of the nuclear factor kappa-light-chain-enhancer of activated B cells (NF-κB) [25]. An experimental study showed that red grape pomace could suppress the expression of inducible nitric oxide synthase (iNOS) and cyclooxygenase-2 (COX-2), highlighting its role in the suppression of inflammation via the NF-κB pathway [26].

As most of the studies are focused on exploring the cardiovascular benefits of red GP, the therapeutic potential of white GP remains largely unexplored, particularly in the context of glucocorticoid-induced hypertension.

Given the rising prevalence of hypertension and the oxidative-inflammatory mechanisms underlying its progression, this study investigates the effects of white GP on blood pressure regulation, oxidative stress, and pro-inflammatory cytokine expression in an experimental model of dexamethasone-induced hypertension. By assessing oxidative stress markers, inflammatory cytokines, and vascular function, the research seeks to elucidate the mechanisms through which GP may counteract the deleterious effects of dexamethasone and contribute to hypertension management.

## 2. Materials and Methods

### 2.1. Ethics Statement

All experiments were conducted in accordance with the Declaration of Helsinki for Animal Studies. The protocol received approval from the Ethics Committee of the “Iuliu Hațieganu” University of Medicine and Pharmacy, Cluj-Napoca as well as from the Veterinary Sanitary and Food Safety Directorate, Cluj-Napoca (Approval No. 255/13.05.2021). The study complied with both national and international guidelines for the care and use of laboratory animals.

### 2.2. Chemicals and Reagents

Dexamethasone Sodium Phosphate (DXA) (Dexamethasone Phosphate Krka 4 mg/mL, Krka d.d. Novo Mesto, Novo Mesto, Slovenia), a standard chemical compound, was purchased from a public pharmacy.

### 2.3. Plant Material

The characterisation of white GP polyphenolic extract was previously detailed [11]. The same extract was used in this experiment. SCDVV Blaj winery (Blaj, Târnave Wine Center, Romania) supplied the white GP that contained the seeds, skins, and stems of *Vitis vinifera* L. white grapes. The cultivars included in the white GP mix were Traminer roz, Sauvignon blanc, Neuburger, Riesling Italian, Fetească regală, Iohaniter, and Muscat ottonel. The grape cultivars were harvested from 12 to 18 September 2019 [11]. The grapes were pressed to yield the white GP, which was then allowed to dry at room temperature. Following drying, the white GP was frozen at −80 °C until it was extracted.

### 2.4. Grape Pomace Polyphenols Extraction and Characterisation

To obtain a rich polyphenol content mixture, equal quantities of 30 g of each white GP cultivar were ground to a fine powder. The powder mix was extracted in 40% ethanol as detailed in the previous work [11]. The ethanolic extract was characterised for its total polyphenol content and phenolic fingerprints. Total polyphenolic content was determined using the Folin–Ciocalteu method [27]. Phenolic compounds fingerprint and composition were determined using Liquid Chromatography-Diode Array Detection–Electro-Spray Ionisation Mass Spectrometry (HPLC-DAD-ESI MS) following the method of Pop et al. [11]. The analysis was conducted using an Agilent 1200 HPLC system coupled with a diode array detector (DAD) and an Agilent 6110 single quadrupole mass spectrometer (Agilent Technologies, Santa Clara, CA, USA). Chromatographic separation was performed at room temperature on an Eclipse XDB C18 column (4.6 × 150 mm, 5 μm particle size) using a binary mobile phase: (A) 0.1% acetic acid/acetonitrile (99:1) in distilled water and (B) 0.1% acetic acid in acetonitrile. The gradient elution profile was as follows: 95% A (0–2 min), linearly reduced to 60% A (2–18 min), then to 10% A (18–20 min), followed by re-equilibration to 95% A (20–26 min). The flow rate was set at 0.5 mL/min, with detection wavelengths at 280 nm and 340 nm. Mass spectrometry was carried out in positive electrospray ionisation (ESI) mode under the following conditions: source temperature 350 °C, nitrogen flow 8 L/min, and capillary voltage 3000 V. Mass spectra were acquired over an m/z range of 100–1000. Data acquisition and analysis were performed using Agilent ChemStation Software (Rev B.04.02 SP1). Compound identification was based on UV spectra, retention times, mass spectra, and literature comparison [11]. Quantification was performed using external calibration curves. Standard solutions of gallic acid, catechin, and rutin were injected in quintuplicate to establish calibration curves for hydroxybenzoic acids (gallic acid equivalent, R^2^ = 0.9978, LOD = 0.35 µg/mL, LOQ = 1.05 µg/mL), flavanols (catechin equivalent, R^2^ = 0.9985, LOD = 0.18 µg/mL, LOQ = 0.55 µg/mL), and flavonols (rutin equivalent, R^2^ = 0.9981, LOD = 0.21 µg/mL, LOQ = 0.64 µg/mL). Thus, the total polyphenol content of the white GP extract rich in polyphenols GP mix extract was 194.18 ± 0.81 mg gallic acid equivalents (GAE)/1 g d.w. of plant material [11]. A high-performance liquid chromatography (HPLC) analysis was previously performed, and 19 phenolic compounds were detected. The extract was predominantly composed of flavanols (66.5% of the total phenolics), followed by flavonols (8.4%) and hydroxybenzoic compounds (7.2%). Additionally, tannins were found in significant concentrations, accounting for 17.7% of the total phenolics. Among the identified compounds, catechin, epicatechin, and procyanidin dimer IV were the most abundant, collectively representing 44.6% of the total phenolics [11].

### 2.5. Animal Grouping and Hypertension Induction

Fifty male Wistar-Bratislava rats (200–250 g) obtained from the Center for Experimental Medicine and Practical Skills of Iuliu Hatieganu University of Medicine and Pharmacy were included in the present study. Rats were maintained in polypropylene cages under standard temperature conditions (22 ± 2 °C), light (12 h light/dark cycles), and humidity. They had unrestricted access to standard food, pellets, and water.

Hypertension (HTN) was induced in groups 2–5 via daily subcutaneous injections of dexamethasone (DEXA) (30 μg/kg body weight (bw). Dexamethasone was already observed to induce HTN in rats at the dose of 30 μg/kg bw [28] or even 20 μg/kg bw [29]. Saline solution (0.5 mL/100 g bw), white GP polyphenols extract (GP), and lisinopril (LIS) were administered orally, by gavage, as previously reported [11]. White grape pomace extract (GP) was administered in two concentrations. The first concentration GP1 was 795 mg polyphenols/kg bw, while the second concentration GP2 was 397.5 mg polyphenols/kg bw. The lisinopril (LIS) concentration was 10 mg/kg bw. All substances were administered daily at the same time. The animals were randomly assigned to five groups (10 rats/group), as follows:

Group 1 (CTRL_S): Control group treated with saline.

Group 2 (CTRL_DEXA): Control group of HTN treated with saline.

Group 3 (DEXA_GP1): Hypertensive group treated with GP1.

Group 4 (DEXA_GP2): Hypertensive group treated with GP2.

Group 5 (DEXA-LIS): Hypertensive group treated with LIS.

All treatments were administered for 16 days. No rat died during follow-up, so the analysis was conducted on all 10 rats in each experimental group.

A flowchart demonstrating the study groups and interventions is presented in Figure 1.

### 2.6. Blood Pressure Measurement

Systolic and diastolic arterial blood pressures were measured in conscious, non-anaesthetised rats, using a tail-cuff plethysmography (Biopac MP36 system, Goleta, CA, USA). The animals were individually restrained in a clear acrylic restrainer at an ambient temperature of 22 ± 2 °C for 15 min. Blood pressures were measured on days 0, 4, 8, 12, and 16 at the same hour every time. Three readings were recorded for each measurement, and the average value was calculated and documented for each group. Biopac Student Lab 3.7.7 software was used to perform the above-mentioned interpretations [30], as shown in Figure 2. The mean arterial pressure (MAP) was calculated as (systolic arterial pressure + 2 × diastolic arterial pressure)/3 [31].

### 2.7. Blood Sampling and Serum Analysis

On day 15, blood samples were collected from the retro-orbital plexus under light anaesthesia (ketamine 20 mg/kg and xylazine 2 mg/kg, intraperitoneally). At the end of the experiment, the rats were euthanised with an overdose of anaesthetics.

Serum concentrations of pro-inflammatory cytokines—tumour necrosis factor-alpha (TNF-α), interleukin-1 beta (IL-1β), and interleukin-6 (IL-6)—were quantified using enzyme-linked immunosorbent assay (ELISA) kits (PeproTech EC, Ltd., London, UK), with a Stat Fax 303 Plus Microstrip Reader (Awareness Technology Inc., Palm City, FL, USA).

Plasma levels of total oxidative status (TOS), total antioxidant capacity (TAC), nitric oxide (NO), malondialdehyde (MDA), and total thiols (THIOL) were measured using a Jasco V-350 UV-VIS spectrophotometer (Jasco International Co., Ltd., Tokyo, Japan). TOS and TAC were determined using fully automated methods developed by Erel, which assess the oxidant–antioxidant balance in biological samples [32,33].

Plasma NO levels were quantified using the method described by Miranda et al., which involves the reduction of nitrate to nitrite by vanadium (III) followed by detection with the Griess reaction [34]. MDA levels, indicative of lipid peroxidation, were measured via the thiobarbituric acid reactive substances assay, as per established protocols [35].

Total thiol content was assessed using Ellman’s reagent (2,2-dithiobisnitrobenzoic acid-DTNB), which reacts with free thiol groups [36].

The oxidative stress index (OSI) was calculated as the ratio of TOS to TAC [37].

### 2.8. Tissue Homogenate

Heart tissues were harvested from all experimental groups immediately post-sacrifice. Each sample was weighed and homogenised in phosphate-buffered saline (PBS) at a 1:4 weight-to-volume ratio using an automated Witeg HG-15D homogeniser (witeg Labortechnik GmbH, Wertheim, Germany). The homogenates were centrifuged at 1500× *g* for 15 min at 4 °C, and the resulting supernatants were collected and stored at −80 °C for subsequent biochemical analyses.

### 2.9. Statistical Analysis

Statistical analysis was conducted using IBM SPSS Statistics version 29.0.0.0 (IBM Corp., Armonk, NY, USA). The Shapiro–Wilk test assessed data distribution. Data with normal distribution was analysed using two-way ANOVA to determine the effects of time on the investigated parameters. Data with non-normal distribution were presented as medians with interquartile ranges (25th–75th percentiles) and were analysed using the Kruskal–Wallis test. Statistical significance was set at *p* < 0.05.

## 3. Results

### 3.1. Effects on Blood Pressure Monitoring

No differences between the groups, in systolic, diastolic, or median blood pressures, were found at the beginning of the experiment (Appendix A).

Dexamethasone administration increased SBP, DBP, and MBP starting from day 4, with higher values being observed at the end of the experiment, on day 16, as shown in Appendix A and Figure 3. The white GP polyphenols extract in DEXA_GP1 reduced SBP and DBP, while lisinopril administration offered good blood pressure control, as shown in Figure 3 and presented in Appendix A.

The two-way ANOVA for repeated measures was used to test the effect of time on rats’ SBP, DBP, and MBP. The test of within-subjects effects indicated that time (*p* = 0.001) and time and group (*p* = 0.001) significantly influenced the SBP, DBP, and MBP of rats included in the study (Figure 3). The Bonferroni test for groups showed that rats treated with dexamethasone had significantly higher levels of SBP, DBP, and MBP (*p* = 0.001) when compared to rats included in the CTRL_S group, except for the DEXA_LIS group, which was not statistically different when compared to the CTRL_S group, indicating the antihypertensive effect of lisinopril. When compared to the CTRL_DEXA group, administration of GP1 showed significantly lower values only for SBP (*p* = 0.001), with no significant difference for DBP or MBP.

### 3.2. Effects on Oxidative Stress Parameters

The present study showed increased serum levels of TOS, MDA, and OSI in rats with HTN, after DEXA administration, as shown in Appendix A. Moreover, dexamethasone administration reduced serum levels of TAC, NO, and Total Thiols (Figure 4; Appendix A).

White GP polyphenol extract with a higher content of polyphenols (GP1) prevented the elevation of serum levels of TOS, MDA, and OSI. The same concentration increased serum levels of TAC, NO, and Total Thiols, the results being significantly higher (*p* < 0.05) than in the group treated with lisinopril (Appendix A; Figure 4).

White GP polyphenols extract with a lower content of polyphenols (GP2) slightly reduced serum levels of MDA and slightly increased the serum levels of NO. This concentration slightly increased the serum levels of NO and significantly elevated TAC and total thiols, when compared to lisinopril (*p* < 0.05) (Appendix A; Figure 4).

In the hypertensive group treated with LIS, the serum levels of TOS, MDA, and OSI were significantly lower (*p* < 0.05) than in the control group of HTN treated with saline (Appendix A; Figure 4). Lisinopril did not influence the levels of NO or Total Thiols (*p* > 0.05).

### 3.3. Effects on Pro-Inflammatory Cytokines

The serum levels of TNF-α were increased in all groups where HTN was induced with dexamethasone, the highest levels being observed in the CTRL_DEXA group (Figure 5).

Both white GP polyphenol extracts prevent the elevation of TNF-α (Figure 5). Similar results were observed for lisinopril (Figure 5).

The serum levels of IL-6 were not influenced (*p* > 0.05) by HTN induction or the treatment administered in all groups (Figure 6). Similar results were also observed for IL-1β in the first four groups (Figure 7).

Heart tissue levels of IL-1β were slightly elevated after DEXA administration, although overall, the tissue levels of the studied cytokines were not influenced (*p* > 0.05) by either the HTN induction or the treatment administered (Figure 5, Figure 6 and Figure 7).

## 4. Discussion

The results of the present study demonstrate that white GP polyphenols extract with a higher content of polyphenols (795 mg polyphenols from GP/kg bw) presents potent antioxidant effects in dexamethasone-induced hypertension, as it prevents the elevation of serum levels of total oxidative stress (TOS), malondialdehyde (MDA), and oxidative stress index (OSI) and increases serum levels of NO, TAC and Total Thiols.

### 4.1. Blood Pressure Monitoring

Dexamethasone induces hypertension due to its role in promoting excessive production of reactive oxygen species (ROS). Increased production of ROS in the vasculature is predominantly caused by the NADPH oxidase pathway [38]. This interaction disrupts the nitric oxide (NO)–redox balance by reducing NO bioavailability, a key vasodilator that the vascular endothelium produces, ultimately contributing to hypertension [39]. Additionally, dexamethasone can inhibit NO synthase gene expression at the transcriptional level, further depleting NO supply and leading to vasoconstriction [40]. Another potential mechanism is dexamethasone-induced elevation of enzymes, such as angiotensin-converting enzyme (ACE), which may significantly increase blood pressure [38]. The hypertensive effects of dexamethasone could be mitigated by counteracting these risk factors through enhanced NO production, inhibition of angiotensin-converting enzyme activity, and improved free radical scavenging. Similar to our study (Figure 4), the dose of Dex (30 μg/kg/day) administered subcutaneously for 16 days was observed to induce hypertension [41].

Grape products are recognised as an excellent source of plant-derived polyphenolic antioxidants, including proanthocyanidins, anthocyanins, flavonols, flavanols, resveratrol, and phenolic acids. Since ACE is a zinc-dependent metalloenzyme, phenolic compounds can bind to its zinc ion, thereby reducing its activity. Consequently, the hypotensive effects of grape products may be attributed to their ability to inhibit ACE activity and enhance prostacyclin levels [42].

In the present study, SBP, DBP, and MBP were controlled after lisinopril administration, as it is an orally active, non-sulfhydryl ACE inhibitor that is not metabolised or bound to protein. Lisinopril produces a smooth, gradual blood pressure reduction without affecting heart rate or cardiovascular reflexes. Its antihypertensive effects begin within 2 h, peak around 6 h, and last for at least 24 h [43].

### 4.2. Oxidative Stress Parameters

Oxidative stress is characterised by an imbalance between the generation and accumulation of ROS in cells and tissues and the biological system’s capacity to neutralise or eliminate these reactive molecules [44]. Total oxidative stress is a pro-oxidant marker often used to estimate the body’s overall oxidation state [45]. Grape pomace polyphenols scavenge free radicals and reduce the accumulation of ROS, thereby decreasing oxidative stress levels [46].

Red grape pomace extract has also been shown to enhance antioxidant defences, particularly TAC [47]. This effect is largely attributed to its flavonoids, resveratrol, and tannins, which scavenge free radicals and prevent oxidative damage [14]. In the present study, white GP mitigated oxidative stress by promoting total thiol production in both concentrations and reducing TAC. Elevated total thiol levels have also been previously observed in other grape pomace extracts [11,48], aligning with the findings of this study. Moreover, the results of the present study show that white GP polyphenol extract with a higher content of polyphenols (GP1) led to a better enhancement of total thiols than lisinopril (Appendix A).

Malondialdehyde (MDA), a key indicator of oxidative stress, is a stable end-product of polyunsaturated fatty acid peroxidation and arachidonic acid metabolism [49]. Previous studies have shown that red wine grape pomace significantly reduces serum MDA levels in models of isoproterenol-induced myocardial ischemia [11,48]. The reduction of MDA levels by grape pomace (GP) reflects decreased lipid peroxidation and oxidative stress, contributing to the protection of cell membranes and supporting cardiovascular health [50].

In the present study, white GP polyphenols extract with a higher content of polyphenols (GP1) significantly increased serum levels of NO, even better than lisinopril. White GP phenolic extracts have shown cardio-protective effects in *ex vitro* models in male Wistar rats, as the rats treated with GP extracts presented relaxation in aortic rings in a dose-dependent manner through the activation of endothelial nitric oxide synthase (eNOS) [51]. Endothelial nitric oxide synthase plays a critical role in regulating and maintaining a healthy cardiovascular system, as decreased production of NO results in an increased susceptibility/risk of developing essential hypertension [52].

Lisinopril, an ACE inhibitor, plays a key role in mitigating oxidative stress in dexamethasone-induced hypertension through several mechanisms. Lisinopril reduces ROS levels by lowering angiotensin II (Ang II), a potent oxidative stress stimulator and enhancing nitric oxide NO bioavailability, which counteracts oxidative damage and promotes vasodilation [53]. In the present study, lisinopril was efficient in controlling blood pressure, even if it presented limited effects on NO serum levels and had no effects on the antioxidant parameters, such as Total Thiols or TAC (Appendix A).

### 4.3. Pro-Inflammatory Cytokines

Hypertension is recognised as a low-grade inflammatory condition marked by the presence of multiple proinflammatory cytokines. Increased levels of Ang II and oxidative stress contribute to the upregulation of TNF-α production [54].

Tumour necrosis factor-α (TNF-α) is a multifunctional cytokine that plays a key role in chronic inflammatory conditions, such as hypertension and diabetes. It has been shown to influence blood pressure regulation in a bidirectional manner. While elevated TNF-α levels can lead to blood pressure reduction, moderate increases have been linked to enhanced sodium chloride retention, contributing to hypertension [55].

The mechanisms through which GP affects TNF-α levels involve the regulation of critical inflammatory pathways. Farina et al. reported that GP extracts suppressed TNF-α-induced NF-κB activation, leading to a reduction in pro-inflammatory cytokine production [56]. Similarly, Nishiumi et al. found that a diet enriched with red GP inhibited NF-κB activation, thereby downregulating the expression of inducible nitric oxide synthase (iNOS) and cyclooxygenase-2 (COX-2) [26]. Lisinopril administration also reduced TNF-α levels, as the use of ACE inhibitors was observed to reduce TNF-α production both in vitro and in vivo [57].

IL-1β belongs to the IL-1 family of interleukins, a group of circulating cytokines implicated in inflammation and disease. Interleukin-1β (IL-1β) is highly relevant in clinical settings due to its elevated levels in hypertensive patients. Given its pivotal role in inflammation, increasing attention has been directed toward understanding its impact on vascular pathology in hypertension. Emerging research indicates that IL-1β not only contributes to the pro-inflammatory response within blood vessels but also affects vascular smooth muscle cell (VSMC) function, phenotype, and vascular remodelling across various forms of hypertension through both inflammatory-dependent and independent pathways [58]. In the present study, the serum levels of IL-1β were not elevated, most probably because dexamethasone is known to have an inhibitory effect on IL-1 production and therefore on IL-1β [59].

Interleukin-6 has been identified in animal studies as a key mediator in angiotensin II (Ang II)-induced hypertension. This pro-inflammatory cytokine is secreted by multiple cell types, including macrophages, endothelial cells, and vascular smooth muscle cells, and plays a central role in sustaining chronic inflammation. IL-6 also stimulates hepatic production of acute-phase proteins, such as C-reactive protein (CRP) [60]. Notably, circulating IL-6 levels may fluctuate based on the context of acute versus chronic inflammatory stimulation [61]. The fact that the present experimental model is a short-time stimulation might be a possible explanation for why the serum levels of IL-6 were not elevated.

Hypertensive heart disease encompasses a range of cardiovascular disorders resulting from prolonged high blood pressure, including left ventricular hypertrophy, heart failure, ischemic heart disease, and arrhythmias. Its development is driven by persistent hypertension, which induces structural and functional changes in the heart over time [62]. The pro-inflammatory cytokines, such as TNF-α, IL-1β, and IL-6, which have been extensively studied for their implication in the pathogenesis of heart failure, were not elevated in the heart tissue, most probably due to a short period of exposure to hypertension [63].

### 4.4. Limitations and Future Directions

Short exposure to hypertension limited our evaluation of the effects of white GP polyphenol extracts on hypertensive heart disease. The lack of evaluation of the oxidative stress parameters and pro-inflammatory cytokines from the aortic wall could be regarded as a potential limitation of the present study.

The pathophysiology of hypertension is multifaceted and involves several mechanisms. These include enhanced salt absorption, which leads to volume expansion, an impaired response of the renin–angiotensin–aldosterone system (RAAS), and increased sympathetic nervous system activation. Together, these alterations contribute to increased total peripheral resistance and elevated afterload, ultimately resulting in the development of hypertension. Future studies should include experimental models focused simultaneously on several pathophysiological mechanisms of arterial hypertension.

## 5. Conclusions

The present study demonstrates that dexamethasone administration induces a significant increase in systolic, diastolic, and mean blood pressure in rats, beginning from day 4 and peaking by day 16, confirming its hypertensive effect. Treatment with white grape pomace extract—particularly the higher concentration (795 mg polyphenols/kg bw)—attenuated the dexamethasone-induced elevation of SBP.

Biochemically, white grape pomace extract in higher concentration significantly reduced oxidative stress markers while increasing antioxidant defence parameters to a greater extent than lisinopril. White grape pomace extract in lower concentration showed a milder antioxidant response, but still surpassed lisinopril, in elevating antioxidant defence parameters. These findings suggest a dose-dependent antioxidant effect of white GP polyphenols.

White grape pomace extract in both concentrations and lisinopril were effective in preventing the rise of tumour necrosis factor-alpha. Interleukin-6 and interleukin-1β levels in serum and heart tissue were not significantly influenced by dexamethasone or any treatment.

To sum up, white grape pomace polyphenol extract exerts protective cardiovascular effects in dexamethasone-induced hypertension by attenuating blood pressure, suppressing oxidative stress, and reducing pro-inflammatory cytokine TNF-α levels, thereby offering a potential natural therapeutic strategy for glucocorticoid-induced hypertension.

## Figures and Tables

**Figure 1 diseases-13-00132-f001:**
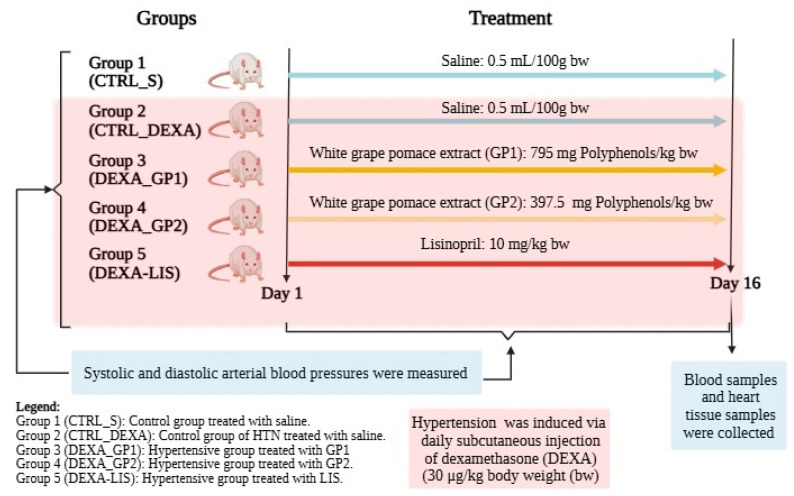
Flowchart demonstrating the study groups and interventions.

**Figure 2 diseases-13-00132-f002:**
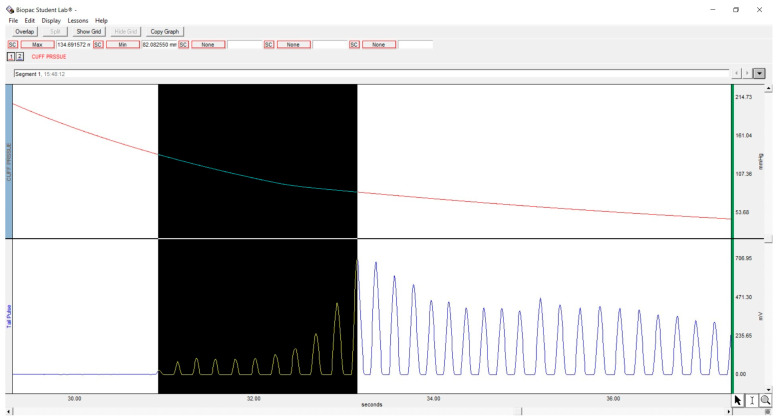
Blood pressure measurement method.

**Figure 3 diseases-13-00132-f003:**
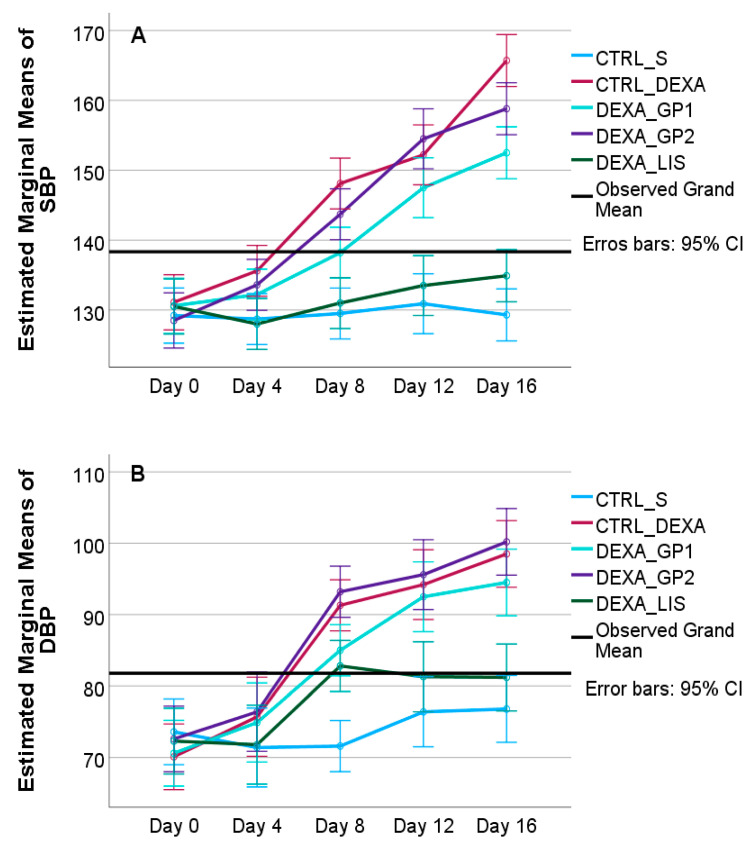
Two-way ANOVA estimated marginal means of SBP (**A**), DBP (**B**), and MBP (**C**) according to the group over time where rats were grouped into control groups, CTRL_S—treated with saline and CTRL_DEXA—treated with dexamethasone, and hypertension groups as follows: DEXA_GP1—treated with white grape pomace concentration 1, DEXA_GP2—treated white grape pomace concentration 2 and DEXA_LIS—treated with lisinopril.

**Figure 4 diseases-13-00132-f004:**
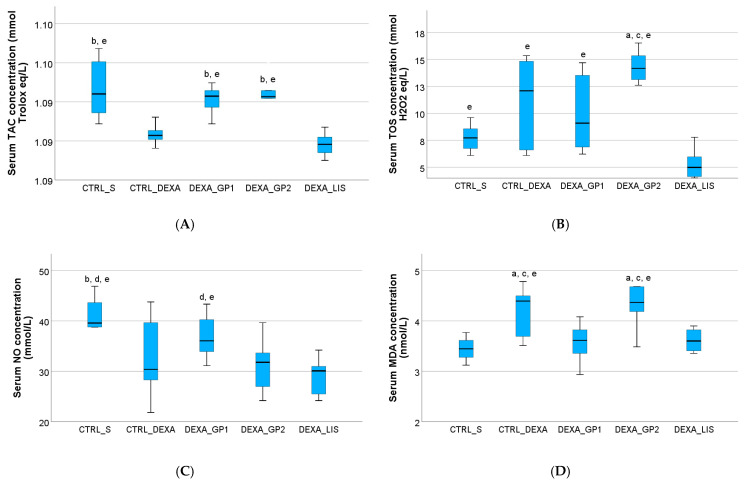
The effect of white grape pomace on serum total antioxidant activity (TAC) (**A**), total oxidative stress (TOS) (**B**), nitric oxide (NO) (**C**), malondialdehyde (MDA) (**D**), total thiols (THIOL) (**E**) and oxidative stress index (OSI) (**F**). Rats were grouped into control groups, CTRL_S—treated with saline and CTRL_DEXA—treated with dexamethasone, and hypertension groups as follows: DEXA_GP1—treated with white grape pomace concentration 1, DEXA_GP2—treated white grape pomace concentration 2 and DEXA_LIS—treated with lisinopril, where ^a^ had *p* < 0.05, versus the CTRL_SF group; ^b^ had *p* < 0.05, versus the CTRL_DEXA group; ^c^ had *p* < 0.05, versus theDEXA_GP1; ^d^ had *p* < 0.05, versus the DEXA_WGP2; and ^e^ had *p* < 0.05, versus the DEXA_LIS, following Kruskal–Wallis Test.

**Figure 5 diseases-13-00132-f005:**
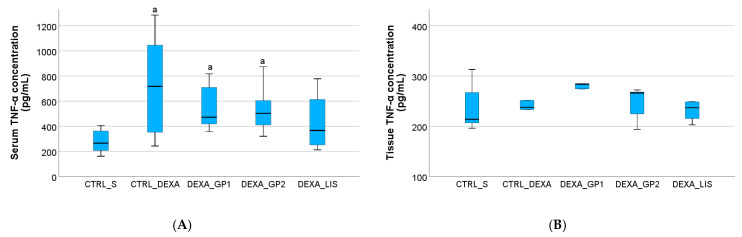
The effect of white grape pomace on tumour necrosis factor alpha (TNF-α) in serum (**A**) and heart tissue (**B**). Rats were grouped into control groups, CTRL_S—treated with saline and CTRL_DEXA—treated with dexamethasone, and hypertension groups as follows: DEXA_GP1—treated with white grape pomace concentration 1, DEXA_GP2—treated white grape pomace concentration 2 and DEXA_LIS—treated with lisinopril, where ^a^ had *p* < 0.05, versus the CTRL_S group following Kruskal–Wallis Test.

**Figure 6 diseases-13-00132-f006:**
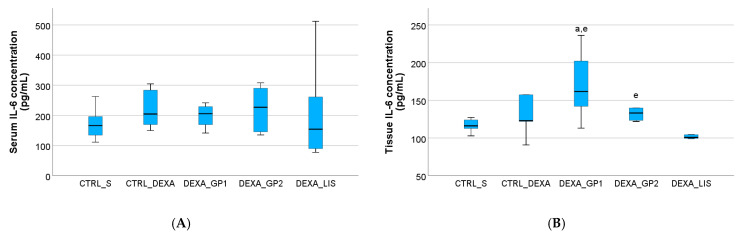
The effect of white grape pomace on interleukin 6 (IL-6) in serum (**A**) and heart tissue (**B**). Rats were grouped into control groups, CTRL_S—treated with saline and CTRL_DEXA—treated with dexamethasone, and hypertension groups as follows: DEXA_GP1—treated with white grape pomace concentration 1, DEXA_GP2—treated white grape pomace concentration 2 and DEXA_LIS—treated with lisinopril, where ^a^ had *p* < 0.05, versus the CTRL_S group and ^e^ had *p* < 0.05, versus the DEXA_LIS group following Kruskal–Wallis Test.

**Figure 7 diseases-13-00132-f007:**
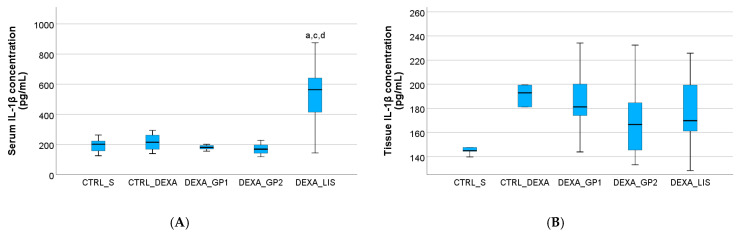
The effect of white grape pomace interleukin 1β (L-1β) in serum (**A**) and heart tissue (**B**). Rats were grouped into control groups, CTRL_S—treated with saline and CTRL_DEXA—treated with dexamethasone, and hypertension groups as follows: DEXA_GP1—treated with white grape pomace concentration 1, DEXA_GP2—treated white grape pomace concentration 2 and DEXA_LIS—treated with lisinopril, where ^a^ had *p* < 0.05, versus the CTRL_S group; ^c^ had *p* < 0.05, versus the DEXA_WGP1; ^d^ had *p* < 0.05, versus the DEXA_GP2 group, following Kruskal–Wallis Test.

## Data Availability

Data are available on request.

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
