# Peer review of "Beneficial Effects of White Grape Pomace in Experimental Dexamethasone-Induced Hypertension"

_diseases, 2025, doi:10.3390/diseases13050132_

Round 1

Reviewer 1 Report

Comments and Suggestions for Authors

The submitted manuscript entitled "Beneficial Effects of White Grape Pomace in Experimental Dexamethasone-Induced Hypertension" represents an additional contribution to the study of the biological activity of polyphenols. The cited literature is correctly presented. The experiments are correct, although some of them require additional clarification.  I would ask the authors for answers.

Did the authors examine the content of polyphenols in individual grape varieties. Are there similarities and differences in the polyphenolic profile of the investigated white grape varieties. What was the idea of ​​making a mixture of the mentioned varieties in the same mass proportion in the further experiment. Such an experiment diminishes the potential application. In real conditions, such a mixture does not exist. The synergism of the polyphenols present must also be taken into account.

The identification and quantification of polyphenols should be significantly clarified. How polyphenols are identified. A molecular ion and probably something else. Based on what standards are they quantified. Were those standards of catechin, and rutin.

Conclusion is quite general and should be reworked a bit. It does not say anything about the effect of different concentrations of polyphenols.

Author Response

First, the authors would like to thank the editor and reviewers for their comments, which have helped improve the manuscript.

The authors have taken into account reviewers’ comments, and the manuscript has been revised. I enclose the answers for reviewers and the revised manuscript entitled " Beneficial Effects of Grape Pomace in Experimental Dexamethasone-Induced Hypertension.

Recommended changes are in red colour in the revised document, to be easily visible.

Comment 1:

Comments and Suggestions for Authors

  • The submitted manuscript entitled "Beneficial Effects of White Grape Pomace in Experimental Dexamethasone-Induced Hypertension" represents an additional contribution to the study of the biological activity of polyphenols. The cited literature is correctly presented. The experiments are correct, although some of them require additional clarification.  I would ask the authors for answers.

Did the authors examine the content of polyphenols in individual grape varieties. Are there similarities and differences in the polyphenolic profile of the investigated white grape varieties. What was the idea of ​​making a mixture of the mentioned varieties in the same mass proportion in the further experiment. Such an experiment diminishes the potential application. In real conditions, such a mixture does not exist. The synergism of the polyphenols present must also be taken into account.

Response:

We thank you for your valuable observations. We examined Traminer roz, Neuburger, Sauvignon blanc, Fetească regală, Riesling Italian, Muscat ottonel, and Iohaniter white grape cultivars. Indeed, there are differences between cultivars. These data were not shown because they will be published separately, in an article that will highlight the differences between grape cultivars according to variety, microclimate and growth conditions. The idea of mixing the grape varieties came after analysing all the varieties cultivated by SCDVV Blaj winery with the aim of valorising the waste. The varieties chosen for the extract had the highest total polyphenol content among the varieties. Our future aim is to be able to valorise the waste as a food supplement. Taking into account that the synergism of the polyphenols is different, we wanted to test and see whether this mix will give the expected results. So far, we have studies that imply single grape varieties as well.  To clarify, we introduced detailed explanations in the material and methods section.

Comment 2:

  • The identification and quantification of polyphenols should be significantly clarified. How polyphenols are identified. A molecular ion and probably something else. Based on what standards are they quantified. Were those standards of catechin, and rutin.

Response:

We thank you for your comment. We have introduced a new section with the missing details.

Comment 3:

  • Conclusion is quite general and should be reworked a bit. It does not say anything about the effect of different concentrations of polyphenols.

Response:

Thank you very much for your suggestion. Conclusions were revised according to your suggestion.

Reviewer 2 Report

Comments and Suggestions for Authors

Present manuscript aimed to investigate the effects of white GP on blood pressure, oxidative stress, and pro-inflammatory cytokines in an experimental model of dexamethasone (DEXA)-induced hypertension (HTN). In generally, the manuscript was well shown; methods were clear. However, some key points limited the acceptance.

  1. In text, results showed that the first concentration GP1 was 795 mg polyphenols/kg bw, while the second concentration GP2 was 397.5 mg polyphenols/kg bw. The effective polyphenols doses were 795 mg polyphenols/kg bw, and 397.5 mg polyphenols/kg bw, respectively. This effective concentration is too high. If this conclusion is correct, the most likely scenario is that polyphenols are a mixture.
  2. The writing English should be improved by native English speaker. I strongly suggest to check the manuscript carefully, especially please check the grammar and the completeness of the sentences once again. And please check the tense of the sentences. There should be consistency throughout the manuscript using past tense.
Comments on the Quality of English Language

Present manuscript aimed to investigate the effects of white GP on blood pressure, oxidative stress, and pro-inflammatory cytokines in an experimental model of dexamethasone (DEXA)-induced hypertension (HTN). In generally, the manuscript was well shown; methods were clear. However, some key points limited the acceptance.

  1. In text, results showed that the first concentration GP1 was 795 mg polyphenols/kg bw, while the second concentration GP2 was 397.5 mg polyphenols/kg bw. The effective polyphenols doses were 795 mg polyphenols/kg bw, and 397.5 mg polyphenols/kg bw, respectively. This effective concentration is too high. If this conclusion is correct, the most likely scenario is that polyphenols are a mixture.
  2. The writing English should be improved by native English speaker. I strongly suggest to check the manuscript carefully, especially please check the grammar and the completeness of the sentences once again. And please check the tense of the sentences. There should be consistency throughout the manuscript using past tense.

Author Response

First, the authors would like to thank the editor and reviewers for their comments, which have helped improve the manuscript.

The authors have taken into account reviewers’ comments, and the manuscript has been revised. I enclose the answers for reviewers and the revised manuscript entitled " Beneficial Effects of Grape Pomace in Experimental Dexamethasone-Induced Hypertension.

Recommended changes are in red colour in the revised document, to be easily visible.

Comment 1:

Comments and Suggestions for Authors

Present manuscript aimed to investigate the effects of white GP on blood pressure, oxidative stress, and pro-inflammatory cytokines in an experimental model of dexamethasone (DEXA)-induced hypertension (HTN). In generally, the manuscript was well shown; methods were clear. However, some key points limited the acceptance.

  1. In text, results showed that the first concentration GP1 was 795 mg polyphenols/kg bw, while the second concentration GP2 was 397.5 mg polyphenols/kg bw. The effective polyphenols doses were 795 mg polyphenols/kg bw, and 397.5 mg polyphenols/kg bw, respectively. This effective concentration is too high. If this conclusion is correct, the most likely scenario is that polyphenols are a mixture.

Response:

We thank the reviewer for their comment. Indeed, the conclusion is correct, and the polyphenols are obtained from a mixture of white grape pomace varieties cultivated by the SCDVV Blaj winery. More details were added in the materials and methods section to clarify these details.

Comment 2:

  1. The writing English should be improved by native English speaker. I strongly suggest to check the manuscript carefully, especially please check the grammar and the completeness of the sentences once again. And please check the tense of the sentences. There should be consistency throughout the manuscript using past tense.

Response:

We thank the reviewer for the comments. The manuscript was revised by a proficient English speaker.

Reviewer 3 Report

Comments and Suggestions for Authors

The authors have dedicated a lot of effort and time in this experimental project which is appreciated. However, unfortunately, the manuscript has several weaknesses that need to be addressed:

1) In your abstract, you have started with "Grape pomace (GP), a byproduct of winemaking, is rich in bioactive polyphenols with potent antioxidant and anti-inflammatory properties. This study aims to investigate the effects of
white GP on blood pressure, oxidative stress, and pro-inflammatory" - this is a very basic start, you need to justify the research gap and novelty of your study here.

2) The introduction needs significant improvement. You need to review the literature and provide an overview of what has been done and their findings with respect to effects of grape bioactives on blood pressure, oxidative stress, and inflammation.

3) Methods - "The characterization of white GP polyphenolic extract was previously detailed (11). The same extract was used in this experiment." You need to provide brief information on how the extract was obtained (i.e., sourcing the GP and extraction technique) - do not expect the reader to back to the previous study for general information. You can mention the general information about the process here and details can be found in the previous article.

4) Figure 1 isn't clear. It need to be improved in terms of resolution and annotation, as well as caption. 

5) It will be useful to add a figure summarising the protocol. 

6) I recommend turning Table 1 into a figure (and keeping the table in the supplementary file) - you figures for the other analysis are great, so you can use the same approach.

7) Results and discussion sections are well written!

8) Conclusion - is very generic, rewrite to be more specific 

Comments on the Quality of English Language

Full review of language is needed. 

Author Response

First, the authors would like to thank the editor and reviewers for their comments, which have helped improve the manuscript.

The authors have taken into account reviewers’ comments, and the manuscript has been revised. I enclose the answers for reviewers and the revised manuscript entitled " Beneficial Effects of Grape Pomace in Experimental Dexamethasone-Induced Hypertension.

Recommended changes are in red colour in the revised document, to be easily visible.

Comment 1:

The authors have dedicated a lot of effort and time in this experimental project which is appreciated. However, unfortunately, the manuscript has several weaknesses that need to be addressed:

  • In your abstract, you have started with "Grape pomace (GP), a byproduct of winemaking, is rich in bioactive polyphenols with potent antioxidant and anti-inflammatory properties. This study aims to investigate the effects of
    white GP on blood pressure, oxidative stress, and pro-inflammatory" - this is a very basic start, you need to justify the research gap and novelty of your study here.

Response:

Thank you for your suggestion! The abstract was reorganised according to your suggestions!

Comment 2:

  • The introduction needs significant improvement. You need to review the literature and provide an overview of what has been done and their findings with respect to effects of grape bioactives on blood pressure, oxidative stress, and inflammation.

Response:

R: Thank you for the valuable feedback. We have reviewed the relevant literature and added a background section in the introduction to provide context and highlight previous findings regarding the effects of grape bioactives on blood pressure, oxidative stress, and inflammation.

Comment 3:

  • Methods - "The characterization of white GP polyphenolic extract was previously detailed (11). The same extract was used in this experiment." You need to provide brief information on how the extract was obtained (i.e., sourcing the GP and extraction technique) - do not expect the reader to go back to the previous study for general information. You can mention the general information about the process here, and details can be found in the previous article.

Response:

We thank you for your comment. We have added two sections that describe plant material sourcing, the extraction, and detection details.

Comment 4:

  • Figure 1 isn't clear. It needs to be improved in terms of resolution and annotation, as well as the caption. 

Response:

We thank you for your suggestion! The figure was revised.

Comment 5:

  • It will be useful to add a figure summarising the protocol. 

Response:

We thank you for your great suggestion! A flowchart was added.

Comment 6:

  • I recommend turning Table 1 into a figure (and keeping the table in the supplementary file) - you figures for the other analysis are great, so you can use the same approach.

Response:

Thank you for your valuable comment. We have introduced the table in the supplementary files and introduced the figures as suggested.

Comment 7:

  • Results and discussion sections are well written!

Response:

Thank you! We appreciate your comment.

Comment 8:

  • Conclusion - is very generic, rewrite to be more specific 

Response:

Thank you very much for your suggestion! Conclusions were revised according to your suggestion.

Round 2

Reviewer 2 Report

Comments and Suggestions for Authors

The manuscript was improved significantly, I suggest accept it after minor revision.

The similarity was 45%, which must be decreased to less than 20%.

Author Response

First, the authors would like to thank the editor and reviewers for their comments, which have helped improve the manuscript.

The authors have taken into account reviewers’ comments, and the manuscript has been revised. I enclose the answers for reviewers and the revised manuscript entitled " Beneficial Effects of Grape Pomace in Experimental Dexamethasone-Induced Hypertension.

Recommended changes are in blue colour in the revised document, to be easily visible.

Comment 1:

The manuscript was improved significantly, I suggest accept it after minor revision.

Response:

We thank the reviewer for the comment and appreciation!

Comment 2:

The similarity was 45%, which must be decreased to less than 20%

Response:

We thank the reviewer for the suggestion. The manuscript was revised, and the suggested paragraphs were rephrased.

Reviewer 3 Report

Comments and Suggestions for Authors

The authors have made all the requested changes.

Author Response

First, the authors would like to thank the editor and reviewers for their comments, which have helped improve the manuscript.

The authors have taken into account reviewers’ comments, and the manuscript has been revised. I enclose the answers for reviewers and the revised manuscript entitled " Beneficial Effects of Grape Pomace in Experimental Dexamethasone-Induced Hypertension.

Recommended changes are in blue colour in the revised document, to be easily visible.

Comment 1:

The authors have made all the requested changes.

Response:

We thank the reviewer for the comment and appreciation!